# DEEP ORIENTATION UNCERTAINTY LEARNING BASED ON A BINGHAM LOSS

**Igor Gilitschenski**[1]**, Roshni Sahoo**[1]**, Wilko Schwarting**[1]**, Alexander Amini**[1]**,
Sertac Karaman**[2]**, Daniela Rus**[1]
[1] Computer Science and Artificial Intelligence Lab, MIT
[2] Laboratory for Information and Decision Systems, MIT
{igilitschenski, rsahoo, wilkos, amini, sertac, rus}@mit.edu

## ABSTRACT

Reasoning about uncertain orientations is one of the core problems in many perception tasks such as object pose estimation or motion estimation. In these scenarios, poor illumination conditions, sensor limitations, or appearance invariance may result in highly uncertain estimates. In this work, we propose a novel learning-based representation for orientation uncertainty. By characterizing uncertainty over unit quaternions with the Bingham distribution, we formulate a loss that naturally captures the antipodal symmetry of the representation. We discuss the interpretability of the learned distribution parameters and demonstrate the feasibility of our approach on several challenging real-world pose estimation tasks involving uncertain orientations.

## 1 INTRODUCTION

Reasoning about uncertain poses and orientations, specifically 3-dimensional (3d) positions and 3-axes orientations, is one of the main inference tasks in computer vision (Sattler et al., 2019), robotics (Glover et al., 2011), aerospace (Crassidis & Markley, 2003), and other fields. Proper representation and estimation of uncertainty is important, e.g. when dealing with structural ambiguities in object pose estimation or coping with sensor corruption.

In vision and robotics tasks, high levels of pose uncertainty may occur due to potentially adversarial conditions that arise in real-world scenarios. A principled approach to uncertainty quantification allows for better execution of planning and situation-awareness tasks such as grasping, tracking, and motion estimation.

When representing uncertainties over poses, the position can be modeled using a Gaussian distribution. This approach is well-motivated by the Central Limit Theorem and widely used in probabilistic deep learning models. However, this paradigm cannot be as easily applied to modeling periodic quantities, such as the orientation of an object. Therefore, Gaussian models become unsuitable particularly in learning regimes involving high uncertainties where one cannot assume local linearity of the underlying space. In this work, we set out to develop a principled probabilistic deep learning approach capable of coping with uncertain orientations.

Currently, most deep learning approaches that predict poses or rigid-body motions suffer from at least one of three drawbacks: 1) they do not model the uncertainty at

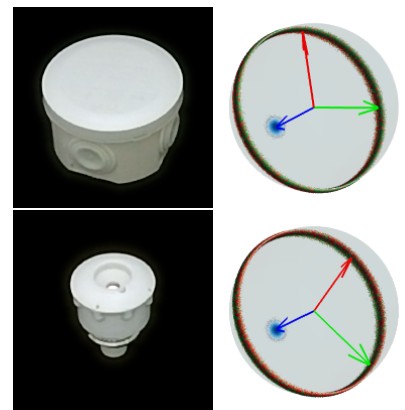

Figure 1: Objects from the T-LESS dataset and the corresponding orientation uncertainty predicted by the model trained on the newly proposed Bingham loss, which is capable of capturing rotational symmetries.

all and merely focus on the accuracy of the predicted pose, 2) they make simplifying assumptions not taking into account that the orientation is defined on a periodic manifold, making the approach

only suitable in low-noise regimes, or 3) even when trying to account for periodicity, no dependency is assumed between the orientation axes and usually an Euler angle-based representation is required. To this point, there are no probabilistic deep learning models for uncertainty of orientations that take the geometry of the underlying domain into account.

In this work, we close this research gap by proposing a probabilistic deep learning model inspired by Directional Statistics (Mardia & Jupp, 1999). We present a loss based on the Bingham distribution (Bingham, 1974), an antipodally symmetric distribution on the sphere. With this loss, we represent uncertain orientations by modeling uncertainty over unit quaternions. Our contributions involve Bingham parameter learning using backpropagation through a Gram-Schmidt method to ensure orthonormalization, efficient approximate evaluation of the normalization constant of the Bingham distribution from a lookup table, and backpropagating through an interpolation scheme during learning. We also discuss interpretability of the Bingham distribution parameters and establish the feasibility of the approach through extensive evaluations.

In summary, this work makes the following contributions: 1) We propose the Bingham loss, a novel loss function for deep learning-based predictions of orientations and their uncertainty. 2) We provide a methodology for making the newly proposed loss and its normalization constant computationally tractable in a deep learning pipeline. 3) We demonstrate multi-modal orientation prediction using a Bingham variant of Mixture Density Networks. 4) We demonstrate how our approach outperforms the state-of-the-art on challenging pose and orientation estimation tasks[1].

## 2 Background: Bingham Distribution for Uncertain Orientations

Unit quaternions are a widely used representation for object orientation in 3d space. They are more compact than rotation matrices, and unlike Euler angles, do not suffer from degeneracies such as Gimbal lock. Additionally, quaternions provide a convenient mathematical notation where the quaternion product, $\mathbf{q}_1 \odot \mathbf{q}_2$, of two unit quaternions $\mathbf{q}_1, \mathbf{q}_2 \in \mathbb{H}_1$ results in a concatenation of the rotations represented by each of the quaternions individually. A full introduction to this representation by given in Kuipers (1999) and notational aspects are discussed by Sommer et al. (2018). In this work, a quaternion $q_1 i + q_2 j + q_3 k + q_4$ will be interpreted as a vector $\mathbf{q} \in \mathbb{R}^4$. It is important to note that the definition of unit quaternions is equivalent to the vector $\mathbf{q}$ being of unit length $||\mathbf{q}|| = 1$. Furthermore, the quaternions $\mathbf{q}$ and $-\mathbf{q}$ represent the same orientation. Therefore, representing uncertain orientations using quaternions requires a probability distribution on the 4d hypersphere that exhibits antipodal symmetry, i.e. for the density function $f(\cdot)$ of this distribution $f(\mathbf{q}) = f(-\mathbf{q})$ has to hold.

A probability distribution exhibiting these properties was proposed by Bingham (1974). It arises by conditioning a zero mean Gaussian to unit length. The Bingham distribution is given in terms of its p.d.f. as $p(\mathbf{x}; \mathbf{M}, \mathbf{Z}) = N(\mathbf{MZM}^\top)^{-1} \exp(\mathbf{x}^\top \mathbf{MZM}^\top \mathbf{x})$ ,where $\mathbf{x} \in \mathbb{R}^4$ with $||\mathbf{x}|| = 1$, $N(\mathbf{MZM}^\top)$ is a normalization constant, $\mathbf{M} \in \mathbb{R}^{4 \times 4}$ orthogonal, and $\mathbf{Z} = \mathrm{diag}(z_1, z_2, z_3, 0) \in \mathbb{R}^{4 \times 4}$ diagonal, with diagonal entries $z_i <= 0$ and the last entry being zero. We use the notation $\mathrm{Bingham}(\mathbf{M}, \mathbf{Z})$. The restriction on the range of the diagonal entries in $\mathbf{Z}$ has numerical and representational convenience reasons. It can be shown that $\mathrm{Bingham}(\mathbf{M}, \mathbf{Z}) = \mathrm{Bingham}(\mathbf{M}, \mathbf{Z} + c\mathbf{I})$ for all $c \in \mathbb{R}$ with $\mathbf{I} \in \mathbb{R}^{4 \times 4}$ denoting the identity matrix. Similarly, changing the order of diagonal entries in $\mathbf{Z}$ has no effect on the distribution as long as the columns in $\mathbf{M}$ are permuted accordingly.

In the definition above, the parameters $\mathbf{M}$ and $\mathbf{Z}$ bear some similarity to the mean and variance of a Gaussian. The density obtains its maxima at $\pm \mathbf{M}_{:,4}$ (the fourth column of $\mathbf{M}$) which can be thought of as a mean orientation respecting the manifold structure. The diagonal entries of $\mathbf{Z}$ can be interpreted as dispersion parameters, and the first three columns of $\mathbf{M}$ can be interpreted as the directions of the dispersion (the Gaussian analog is the orientation of the covariance ellipsoid). Bingham distributions allow for representation of uniform priors over individual axes or even the entire space, making them superior to Gaussians in any of the usual orientation representations.

---

[1]Code available at `https://github.com/igilitschenski/deep_bingham`

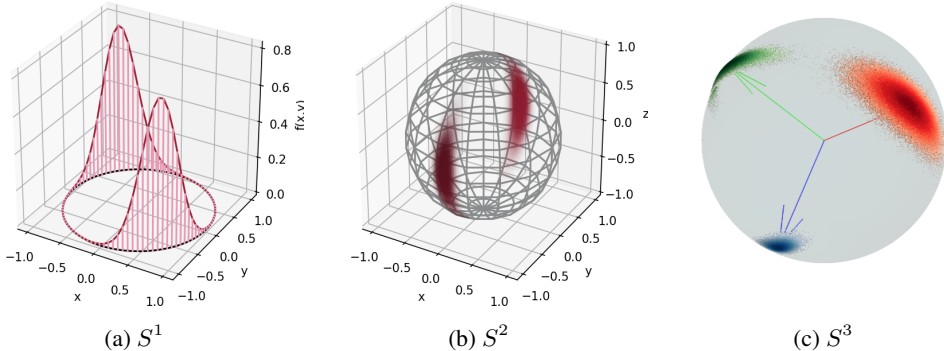

(a) $S^1$            (b) $S^2$            (c) $S^3$

Figure 2: Densities of the Bingham distribution represented for different dimensionality. For the circular case (a), the density is shown as a function of unit vectors on the plane. For the spherical case (b), it is shown as a heatmap on a 3d unit sphere. For the 4d case (c), which is of our particular interest, we visualize the mode of the Bingham in terms of the coordinate system orientation represented by the corresponding quaternion. Then, we draw samples from the distribution and visualize each sample as a potential coordinate arrow endpoint for each axis (i.e. each sample drawn from the Bingham distribution is represented by three points in the plot). This representation allows us to simultaneously represent the orientation and the corresponding uncertainty.

One of the main challenges of using the Bingham distribution is the computation of its normalization constant

$$N(\mathbf{M}\mathbf{Z}\mathbf{M}^\top) = \int_{||q||=1} \exp(\mathbf{q}^\top \mathbf{M}\mathbf{Z}\mathbf{M}^\top \mathbf{q}) \, \mathrm{d}\mathbf{q} \, ,$$

which is a Hypergeometric function of matrix argument (Herz, 1955). Evaluating these functions imposes a high computational burden and is still an area of active research (Koev & Edelman, 2006; Kume et al., 2013; Koyama et al., 2014; Kume & Sei, 2018). Using the transformation theorem and the fact that $\mathbf{M}$ is orthogonal, the normalization constant can be simplified as $N(\mathbf{M}\mathbf{Z}\mathbf{M}^\top) = N(\mathbf{Z})$, making it merely a function of the three parameters $z_i$ ($i = 1, 2, 3$) and motivating the use of precomputed lookup tables in practice.

Furthermore, to make the uncertainty of a Bingham Distribution more interpretable in practice, we propose the use of Expected Absolute Angular Deviation (EAAD) which is defined as

$$\mathrm{EAAD}(\mathbf{Z}) = \int_{||q||=1} \theta(\mathbf{q}, \mathbf{e}) \cdot p(\mathbf{q}; \, \mathbf{I}, \mathbf{Z}) \, \mathrm{d}\mathbf{q} \, ,$$

where $p(\cdot)$ is the $\mathrm{Bingham}(\mathbf{I}, \mathbf{Z})$ density, $\mathbf{I}$ is the identity matrix, $\mathbf{e} = [0, 0, 0, 1]$ is the vector corresponding to the unit quaternion representing the identity and $\theta(\mathbf{q}, \mathbf{e}) = 2 \cdot \arccos(|\langle \mathbf{q}, \mathbf{e} \rangle|)$ denotes the angular distance between $\mathbf{q}$ and $\mathbf{e}$. The EAAD describes the expected angular deviation from the "mean" orientation. It can be loosely thought of as the orientation counterpart to the standard deviation in Euclidean space. For the same reason as in the normalization constant, the EAAD computation does not involve the parameter $\mathbf{M}$.

## 3   DEEP ORIENTATION UNCERTAINTY LEARNING

The Bingham distribution is the main component of the proposed probabilistic framework for representing deep learned uncertain orientations. Drawing inspiration from Mixture Density Networks (Bishop, 1994), we propose using the Bingham distribution's negative log-likelihood as a loss function

$$L(\mathbf{y}, \mathbf{M}, \mathbf{Z}) = -\log p(\mathbf{y}; \, \mathbf{M}, \mathbf{Z}) = -\mathbf{y}^\top \mathbf{M}\mathbf{Z}\mathbf{M}^\top \mathbf{y} + \log N(\mathbf{Z}) \, ,$$

with $\mathbf{M}$, $\mathbf{Z}$ as defined above and $\mathbf{y}$ being the orientation label given in the training data. We use a neural network to learn $\mathbf{M}$ and $\mathbf{Z}$, end-to-end, directly from the input data (e.g. RGB images). From

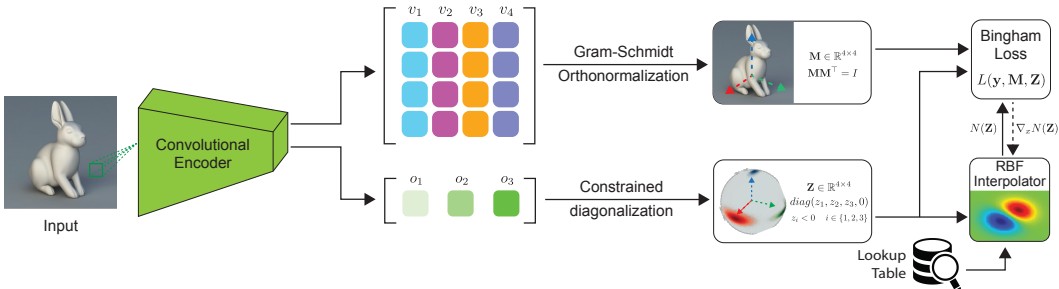

Figure 3: The proposed orientation uncertainty estimation pipeline predicts the parameters of a Bingham distribution for representing uncertain unit quaternions. Backpropagation through an interpolator and use of a lookup table allows for avoiding evaluations of the computationally expensive Bingham normalization constant.

this prediction, the point estimate of $\mathbf{y}$ is obtained as $\hat{\mathbf{y}} = \mathbf{M}_{:,4}$ as the last column corresponds to the highest diagonal entry of $\mathbf{Z}$ and thus represents one of the modes of the distribution (the other being $-\hat{\mathbf{y}}$ due to antipodal symmetry).

No costly evaluation of the normalization constant is required and no major computational challenges arise in the special case where the dispersion parameter $\mathbf{Z}$ is known and not predicted by a neural network. However, as our goal is the modeling of uncertainty, we propose methods for modeling $\mathbf{M}$ and $\mathbf{Z}$ as well as backpropagating through $N(\mathbf{Z})$.

### 3.1 MODELING OF DISTRIBUTION PARAMETERS

In order to obtain predictions $\hat{\mathbf{M}}$ and $\hat{\mathbf{Z}}$, we require a 19 dimensional output ($\mathbf{o} \in \mathbb{R}^{19}$) of the predictor network (3 outputs for $\mathbf{Z}$, 16 outputs for $\mathbf{M}$). On its own, these outputs do not satisfy the above-mentioned constraints on the Bingham distribution parameters. Thus, we define the differentiable transforms $T_{\mathbf{M}} : \mathbb{R}^{16} \to \mathbb{R}^{4 \times 4}$ and $T_{\mathbf{Z}} : \mathbb{R}^{3} \to \mathbb{R}^{4 \times 4}$ that transform these outputs such that the constraints are satisfied.

The transform $T_{\mathbf{Z}}$ is obtained as $T_{\mathbf{Z}}(o_1, o_2, o_3) = \mathrm{diag}(\hat{z}_1, \hat{z}_2, \hat{z}_3, 0)$ with $\hat{z}_i = -\exp(o_i)$. For computing $\hat{\mathbf{M}}$, we first subdivide $o_4, \ldots, o_{19}$ into four vectors $\mathbf{v}_i \in \mathbb{R}^4$ ($i = 1, \ldots, 4$). Then, we apply the Gram-Schmidt orthonormalization method to these vectors according to $\hat{\mathbf{m}}_i = \mathrm{Normalize}(\mathbf{v}_i - \sum_{k=1}^{i-1} \langle \hat{\mathbf{m}}_k, \mathbf{v}_i \rangle \cdot \hat{\mathbf{m}}_k)$ with $i \in \{1, 2, 3, 4\}$ and $\mathrm{Normalize}(\mathbf{x}) = \mathbf{x}/||\mathbf{x}||$. Finally, the prediction $\hat{\mathbf{M}}$ is obtained as $T_{\mathbf{M}}(o_3, \ldots, o_{19}) = [\hat{\mathbf{m}}_1, \ldots, \hat{\mathbf{m}}_4]$, and $\hat{\mathbf{M}}$ is orthogonal by construction.

### 3.2 BACKPROPAGATION THROUGH THE BINGHAM NORMALIZATION CONSTANT

As mentioned earlier, computation of the Bingham normalization constant is numerically burdensome. This is also true for its derivatives which can be shown to be proportional to the normalization constant of Bingham distributions of higher dimension (Kume & Wood, 2007). A forward-backward pass for one single data point requires 4 evaluations of hypergeometric functions of matrix argument.

We avoid this by precomputing a lookup table for $N(\mathbf{Z})$ at L different locations $\mathbf{t}_i$ (with $\mathbf{Z}_i = \mathrm{diag}([\mathbf{t}_i^\top, 0])$. This table is then used to build an interpolator $f_N(\mathbf{z}) = \sum_{i-1}^{L} w_i \phi(||\mathbf{z} - \mathbf{t}_i||)$ with $\mathbf{z} \in \mathbb{R}^3$ and $\phi$ denoting a radial basis function. The weights $w_i$ can also be precomputed during generation of the interpolator. Thus, we can approximate $N(\mathbf{Z}) \approx f_N(\mathbf{z})$ and $\nabla_{\mathbf{z}} N(\mathbf{Z}) \approx \nabla_{\mathbf{z}} f_N(\mathbf{z})$. To the best of our knowledge, this is the first time that a lookup table based interpolation mechanism has been included in the computation graph of a neural network.

### 3.3 MULTI-MODAL PREDICTION

A Bingham variant of Mixture Density Networks can be used to obtain multi-modal predictions. However, MDNs are hard to train even in the Gaussian case. Following the discussion in Makansi

et al. (2019), we separate the training in two stages. In the first stage, we only learn to predict $\mathbf{M}$ and assume the dispersion to be fixed with $\mathbf{Z} = \mathrm{diag}(-a, -a, -a, 0)$. In practice $a \in \mathbb{R}^+$ can usually be set to 1 as it merely scales the cost term. In the second stage, we train to predict $\mathbf{M}$ and $\mathbf{Z}$ jointly. Our evaluation will show that in high uncertainty regimes, this training method is also helpful for the unimodal case.

## 4 EXPERIMENTS

In this section we evaluate the proposed Bingham loss on its ability to learn calibrated uncertainty estimates for orientations. This goes beyond comparing point estimates of orientations; we evaluate how well the estimated *distribution* of orientations can explain the data. We will also show that the Bingham distribution representation is capable of capturing ambiguity and uncertainty in SO(3) better than state-of-the-art approaches.

We investigate characteristics and behaviors by training neural networks on two head-pose datasets, IDIAP (Odobez, 2003) and UPNA (Ariz et al., 2016), as well as the object pose dataset T-LESS (Hodaň et al., 2017). We show the capability of calibrated uncertainty estimation by applying artificial label-noise to IDIAP and UPNA and observing that the Bingham parametrization allows for accurate prediction of uncertainty. In addition to calibrated uncertainty estimation, we demonstrate advanced capabilities in the face of object orientation ambiguity on the T-LESS dataset by visualizing the predicted distributions for different orientation ambiguous objects, e.g. symmetric, and comparing to objects with clear orientation.

### 4.1 ARCHITECTURE AND EXPERIMENTAL SETUP

We seek to estimate the Bingham distribution parameters directly from image data. Our pipeline is shown in Figure 3 and begins by passing an image input to a convolutional encoder, in our case a standard ResNet-18 network followed by a fully connected layer, populating the entries of $o_1, o_2, o_3$ and $v_1, v_2, v_3, v_4$. Subsequently, $\mathbf{Z}$ is computed by constrained diagonalization of $o_1, o_2, o_3$, and Gram-Schmidt orthonormalization of $v_1, v_2, v_3, v_4$ yields $\mathbf{M}$, as described in Section 3.1. To evaluate the Bingham loss, the normalizer $N(\mathbf{Z})$ needs to be queried from the RBF lookup table, Section 3.2. Differentiation of the interpolator via finite differences enables us to back-propagate through the entire pipeline. All models were implemented in PyTorch and optimized with the Adam optimizer.

We create the lookup table by numerical integration. More precisely, we use Scipy's `tplquad` method to compute a triple integral for each $\mathbf{Z}$ in the table. We set the relative error tolerance to 1e-3 and the absolute error tolerance to 1e-7. The actual computed integral is

$$N(\mathbf{Z}) = \int_0^{2\pi} \int_0^{\pi} \int_0^{\pi} \exp\left(t(\phi_1, \phi_2, \phi_3)^\top \mathbf{Z}\, t(\phi_1, \phi_2, \phi_3)\right) \cdot \sin(\phi_1)^2 \cdot \sin(\phi_2)\, \mathrm{d}\phi_1\, \mathrm{d}\phi_2\, \mathrm{d}\phi_3 \,,$$

with

$$t(\phi_1, \phi_2, \phi_3) = \begin{bmatrix} \sin(\phi_1) \cdot \sin(\phi_2) \cdot \sin(\phi_3) \\ \sin(\phi_1) \cdot \sin(\phi_2) \cdot \cos(\phi_3) \\ \sin(\phi_1) \cdot \cos(\phi_2) \\ \cos(\phi_1) \end{bmatrix}$$

to account for a transformation of coordinates from unit quaternions to 4d spherical coordinates. Because we use the Bingham log likelihood as our optimization objective, we compute the logarithm before the interpolation to avoid failure at locations where the interpolator wrongly outputs negative values.

### 4.2 BASELINES

We compare our work with the approach proposed by Prokudin et al. (2018). It also uses a loss based on directional statistics, specifically the Von Mises distribution. The Von Mises distribution can be thought of as a circular analog of the Normal distribution. In order to apply this approach to our setting, orientations are modeled with Euler angles. The loss then consists of the sum of log-likelihoods for each angle. While this approach can properly account for periodicity of the

underlying data, we expect it to fail in cases where the underlying uncertainty is not axis aligned because it does not account for dependencies between uncertain rotation axes.

Furthermore, we also evaluate several different representations of the parameter matrix $\mathbf{M}$. We consider the classical Gram-Schmidt (CGS), modified Gram-Schmidt (MGS), and the matrix representation of the quaternion (QM) used by Birdal et al. (2018). Finally, we also include two non-probabilistic orientation prediction baselines. The first one is based on a Mean Square Error (MSE) between the predicted and ground truth quaternion. The second one is based on a cosine loss applied to each angle's biternion as discussed by Prokudin et al. (2018).

## 4.3 EVALUATION METRICS

To evaluate error metrics over predicted orientations, it is unsuitable to compute the RMSE over angles, since it does not sufficiently consider the spherical nature of the underlying data. Instead, we make use of the Mean Absolute Angular Deviation (MAAD) which has also been used by Prokudin et al. (2018). It is based on the angular distance between two angles defined above. We also compute the EAAD to assess the quality of the results. Additionally, the difference between EAAD and MAAD serves as an indicator of the quality of the predicted uncertainty. The acceptable difference in practice is application dependent. For the cases of the Von Mises distribution parameters, EAAD computation is carried out in a similar way as for the Bingham defined above. EAAD is calculated over the learned dispersion parameters for each example and averaged. The quality of the respective model is measured in terms of log-likelihood to indicate the goodness of an individual fit. For MDNs, we additionally report a Mean Minimum Absolute Angular Deviation (MMAAD), which uses the component closest to ground-truth for absolute angular deviation computation. The MAAD and EAAD for MDNs are computed in a per-component fashion and then weighted using the predicted mixture weights.

## 4.4 CALIBRATED UNCERTAINTY ESTIMATION

We evaluate the distribution fit on the head pose datasets UPNA and IDIAP, which consist of head images from a video of several people inside a room. Each image is annotated with head orientation given by pan, tilt and roll angles. We use these datasets as they provide accurate labels and allow for carrying out experiments involving artificial label noise.

The results on the raw dataset are shown in Table 1. They demonstrate that the general performance for point estimates, indicated by MAAD, of the Bingham distribution remains on a similar level as the Von Mises distribution and the non-probabilistic approaches. In this setting, most motions of the subjects' heads are aligned with the gravity axis allowing both distributions to successfully capture

|  | UPNA | | | IDIAP | | |
|---|---|---|---|---|---|---|
|  | EAAD | MAAD | LL | EAAD | MAAD | LL |
| BD-CGS | 0.10 | 0.11 | 4.70 | 0.10 | 0.09 | 4.49 |
| BD-MGS | 0.10 | 0.13 | 3.87 | 0.10 | 0.10 | 4.58 |
| BD-QM | 0.10 | 0.16 | 0.31 | 0.10 | 0.09 | 4.74 |
| VM | 0.13 | 0.11 | 3.69 | 0.12 | 0.09 | 2.08 |
| MSE | - | 0.12 | - | - | 0.10 | - |
| Cosine | - | 0.12 | - | - | 0.10 | - |

Table 1: Bingham (BD), Von Mises (VM), Mean Square Error (MSE), and cosine based loss prediction performance on raw UPNA and IDIAP datasets.

the noise. However, the Bingham still attains a higher log-likelihood and a smaller gap between MAAD and EAAD. Similarly, the parametrization of the concentration matrix $\mathbf{M}$ has a relatively small impact on the estimation performance. Although MGS has stronger robustness guarantees than CGS (the latter has a quadratic dependency on the condition number of the input matrix, see Giraud et al. (2005) for a discussion of both), the condition of the input is not poor enough to impact performance. While the quaternion matrix approach is easier to train, it also loses some of the expressiveness of the Bingham distribution because the underlying mapping (from quaternions to the space of orthogonal matrices) is not surjective.

To estimate how well the predicted uncertainties are calibrated, we add artificial noise by drawing random perturbations from the Bingham distribution with varying $z_1, z_2,$ and $z_3$ parameters and applying them to the quaternion labels before training. Both UPNA and IDIAP contain negligible

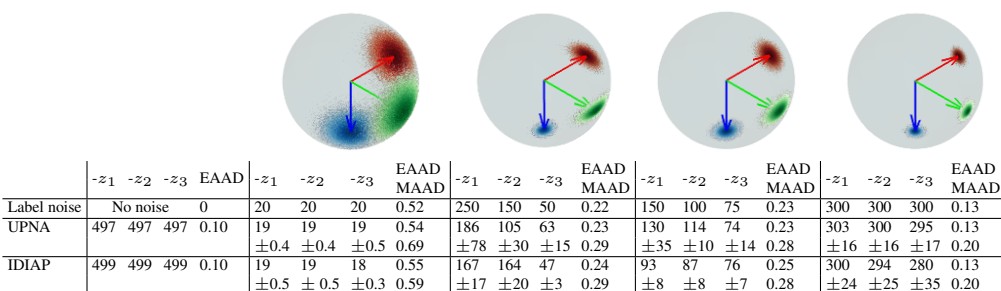

| Label noise | | -z1 | -z2 | -z3 | EAAD | -z1 | -z2 | -z3 | EAAD MAAD | -z1 | -z2 | -z3 | EAAD MAAD | -z1 | -z2 | -z3 | EAAD MAAD | -z1 | -z2 | -z3 | EAAD MAAD |
|---|---|---|---|---|---|---|---|---|---|---|---|---|---|---|---|---|---|---|---|---|---|
| Label noise | No noise | | | | 0 | 20 | 20 | 20 | 0.52 | 250 | 150 | 50 | 0.22 | 150 | 100 | 75 | 0.23 | 300 | 300 | 300 | 0.13 |
| UPNA | | 497 | 497 | 497 | 0.10 | 19 | 19 | 19 | 0.54 | 186 | 105 | 63 | 0.23 | 130 | 114 | 74 | 0.23 | 303 | 300 | 295 | 0.13 |
| | | | | | | $\pm0.4$ | $\pm0.4$ | $\pm0.5$ | 0.69 | $\pm78$ | $\pm30$ | $\pm15$ | 0.29 | $\pm35$ | $\pm10$ | $\pm14$ | 0.28 | $\pm16$ | $\pm16$ | $\pm17$ | 0.20 |
| IDIAP | | 499 | 499 | 499 | 0.10 | 19 | 19 | 18 | 0.55 | 167 | 164 | 47 | 0.24 | 93 | 87 | 76 | 0.25 | 300 | 294 | 280 | 0.13 |
| | | | | | | $\pm0.5$ | $\pm0.5$ | $\pm0.3$ | 0.59 | $\pm17$ | $\pm20$ | $\pm3$ | 0.29 | $\pm8$ | $\pm8$ | $\pm7$ | 0.28 | $\pm24$ | $\pm25$ | $\pm35$ | 0.20 |

Table 2: Testing accuracy of uncertainty calibration. Prior to training, we perturb the labels with noise sampled from the Bingham distribution with $\mathbf{M}$ equal to the identity and varying $z_1, z_2, z_3$. The figures represent the different noise distributions.

amounts of noise, so the dispersion of the noise distribution should be captured by the learned $\mathbf{Z}$ to high accuracy. An evaluation of uncertainty and label noise is shown in Table 2. For the case of no noise, the Bingham uncertainty parameters approximate the highest certainty levels represented in the lookup table. Thus, the maximum and minimum values in the lookup table automatically become the bounds of what certainty levels can be represented by the proposed loss. When noise is applied to the training labels, the learned uncertainty parameters closely match the dispersion of label noise, so the predicted EAAD accurately captures the EAAD corresponding to the dispersion of the label noise distribution. We note that the MAAD is slightly higher than the true and estimated EAAD values. This overconfidence effect is typical in probabilistic deep learning and also arises when predicting the parameters of a Gaussian (Amini et al., 2019). In addition, we evaluated a scenario where the noise is newly sampled and applied to the true labels in each iteration (rather than corrupting the labels with the sampled noise prior to training). In this scenario, the EAAD computed from the learned dispersion parameters, the true EAAD, and the MAAD are approximately equal in value. While this scenario is less realistic in practice (and thus not visualized), it provides further evidence for representational consistency of the loss.

## 4.5 HANDLING AMBIGUOUS DATA

We use the T-LESS dataset for evaluating the proposed model using ambiguous data. It contains images of 30 different textureless objects taken from different cameras. We use the Kinect RGB single-object images all of which are split into training, test, and validation sets. At a coarse scale most of the objects in the dataset exhibit rotational or other symmetries. At a finer scale some of these ambiguities disappear due to smaller structures. On the one hand, we expect those to be more challenging to learn. On the other hand, capturing these structures allows for very precise orientation estimation. To be able to disregard these structures, we create a variant of T-LESS where we add blur to each image using a uniform $10\text{px} \times 10\text{px}$ kernel.

We carry out two sets of experiments. In the first set of experiments, we train orientation estimation models for 5 epochs using the Bingham loss (BD-5) and the Von Mises loss (VM-5) on the blurred and original set of images. This allows to investigate the uncertainty estimation properties before the network captures the finer grained structures. In the second set of experiments, we use the original set of images to evaluate multi-modal orientation prediction using the two-stage training approach for models with 1

| Method | Log-likelihood | MAAD | EAAD |
|---|---|---|---|
| VM-5 | -0.12 | 0.48 | 0.33 |
| BD-5 | 2.82 | 1.57 | 1.58 |
| VM-5 w. blur | -0.03 | 0.56 | 0.44 |
| BD-5 w. blur | 2.71 | 1.59 | 1.58 |

Table 3: Results on the T-LESS dataset in the high uncertainty regime.

(BD-MDN-1), 2 (BD-MDN-2), and 4 (BD-MDN-4) mixture components. Each stage is carried out for 30 epochs. The comparison methods use Von Mises (VM), Mean Square Error (MSE), and Cosine losses with an overall training duration of 60 epochs (or until convergence if that is earlier).

The results for the first set of experiments are visualized in Table 3. As expected, both approaches are on average far off in terms of the true orientation. While Von Mises performs better on the MAAD, we observe that there is a larger difference between the MAAD and EAAD values for the

Von Mises distribution than the Bingham distribution. This indicates that the uncertainty estimates of the Von Mises distribution may be overconfident. On the other hand the Bingham distribution better captures the uncertainty over individual axes. One interesting insight is that allowing for uniform distributions over individual non-aligned periodic axes can make it hard for the learning method to pick up on the proper pose and thus may require pre-training on the pure pose estimation task in such regimes.

In the second set of experiments, as visualized in Table 4, we use this training strategy for all Bingham MDN models resulting in robust convergence behavior. However, the unimodal Bingham (BD-MDN-1) converges slower than Von Mises (VM) thus achieving a higher MAAD, which is adequately captured by the Bingham's EAAD. For multiple mixture components, we obtain a very low MAAD and can observe again the phenomenon of the lookup table limitations in the EAAD. Thus, the

| Method | Log-likelihood | MAAD | MMAAD | EAAD |
|--------|----------------|------|-------|------|
| VM | 3.73 | 0.10 | - | 0.17 |
| BD-MDN-1 | 5.00 | 0.20 | - | 0.21 |
| BD-MDN-2 | 6.17 | 0.07 | 0.06 | 0.12 |
| BD-MDN-4 | 6.19 | 0.06 | 0.05 | 0.10 |
| MSE | - | 0.22 | - | - |
| Cosine | - | 0.10 | - | - |

Table 4: Results on the T-LESS dataset involving multi modal prediction.

MAAD achieved during the first training stage can not only be used for inspecting the network's accuracy but also for determining the minimum $\mathbf{Z}$ parameter values stored in the lookup table. Another interesting phenomenon can be observed in the EAAD and MAAD of the VM loss. As the representation required by Von Mises assumes that each axis is independent, EAAD is computed per rotation axis. This results in an overapproximation of the uncertainty overall. For the non-probabilistic losses, the cosine loss achieves better performance which is probably due to better consideration of the underlying geometry. In summary, while the proposed Bingham loss shares the general challenges of training Mixture Density Networks, it better captures the underlying noise structure by explicitly modeling dependencies between rotation axes.

# 5 DISCUSSION AND RELATED WORK

Quantifying and representing uncertainty by and in neural networks has been a subject of extensive research initially focused on modeling probability distribution parameters (Nix & Weigend, 1994) and mixture distributions (Bishop, 1994) as neural network outputs. More recent approaches focus on improving understanding of the underlying uncertainties (Kendall & Gal, 2017), providing scalable techniques for estimating predictive uncertainty (Lakshminarayanan et al., 2017), and stabilizing training to avoid mode collapse (Makansi et al., 2019). The present work is orthogonal to these approaches in the sense that it focuses on proper modeling of the underlying geometric domain and coping with a computationally demanding normalization constant.

Handling of poses and orientations has been extensively studied in the context of Bayesian filtering for applications such as spacecraft attitude estimation (Crassidis & Markley, 2003) and ego-motion estimation (Bloesch et al., 2015), where one can often assume the underlying uncertainties to be small. This allows for leveraging local-linearity and using the Gaussian distribution. Recently, methods based on directional statistics enabled modeling of high uncertainty levels for inferring orientations (Gilitschenski et al., 2016) and full poses (Glover et al., 2011; Glover & Kaelbling, 2014; Srivatsan et al., 2016) by using the Bingham distribution. Drawing inspiration from these results, this work extends the applicability of these approaches to probabilistic deep learning models.

Particularly in computer vision, deep learning has been applied to spherical regression and pose estimation problems (Liao et al., 2019; Huang et al., 2018). These applications involve inferring object (Brachmann et al., 2014; Hodaň et al., 2018; Li et al., 2018b;a; Manhardt et al., 2019; Sundermeyer et al., 2018; Tekin et al., 2018; Wang et al., 2019b;a), body (Yang et al., 2019), and camera poses (Clark et al., 2017; Sattler et al., 2019; Wang et al., 2017; 2018). In all of these scenarios there is a multitude of sources for potentially high uncertainties such as the use of low-resolution data (e.g. tracking pose of distant pedestrians), absence of textures (e.g. when operating on depth data), or motion blur (e.g. due to high speeds in ego-motion estimation). However, most of the existing approaches merely focus on inferring the pose but do not account for the underlying uncertainty.

The representation proposed in our work closes this gap by allowing for neural networks to output well-calibrated orientation uncertainty estimates.

Only a few approaches consider modeling the uncertainty of orientations for deep learning based pose estimation. *PoseRBPF* by Deng et al. (2019) discretizes the orientation space into over 190 000 bins and learns a codebook to allow for tractable inference. In contrast to that approach, we do not require an a priori discretization and can directly obtain interpretable estimates. Similarly to us, Prokudin et al. (2018) propose a loss based on directional statistics. By making use of the Von Mises distribution, their work can properly account for periodicity of circular data. However, as we have shown in our evaluations, this approach cannot properly account for dependencies between different axes and thus, struggles when the underlying uncertainty is not axis aligned.

## 6 CONCLUSION

In this work, we introduced the Bingham loss, a loss function based on the Bingham distribution that enables neural networks to predict uncertainty over unit quaternions and thus uncertain orientations. This allows for using (rotation-)symmetric objects and ambiguous sensor data in the context of pose and orientation estimation. In addition, we demonstrate how to cope with intractable likelihoods in deep learning pipelines by using non-linear interpolation and lookup tables as part of the computation graph.

The presented approach is directly usable in existing probabilistic deep learning techniques. Moreover, we demonstrate its applicability for mixture density models. The choice of parametrization remains one of the main design decisions in pose and orientation estimation pipelines. Our work supports the case for using quaternions over other parametrizations for deep learning. It also motivates further research on how to properly model dependencies between uncertain periodic and non-periodic quantities.

### ACKNOWLEDGMENTS

This work was supported in part by NSF Grant 1723943, the Office of Naval Research (ONR) Grant N00014-18-1-2830, and Toyota Research Institute (TRI). This article solely reflects the opinions and conclusions of its authors and not TRI, Toyota, or any other Toyota entity. Their support is gratefully acknowledged.

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
