# OpenReview forum: "Deep Orientation Uncertainty Learning based on a Bingham Loss"
_ICLR.cc/2020/Conference — Accept (Poster)_

### Official Review · AnonReviewer3 · 2019-10-28
**Official Blind Review #2513**

**Rating:** 6

**Review:**

The paper proposes a Brigham loss (based on the Brigham distribution) to model the uncertainty of orientations (an important factor for pose estimation and other tasks). This distribution has the necessary characteristics required to represent orientation uncertainty using quaternions (one way to represent object orientation in 3D) such as antipodal symmetry. The authors propose various additions such as using precomputed lookup tables to represent a simplified version of the normalization constant (to make it computationally tractable), and the use of Expected Absolute Angular Deviation (EAAD) to make the uncertainty of the Bingham distribution more interpretable.

+Uncertainty quantification of neural networks is an important problem that I believe should gain more attention so I am happy to see papers such as this one.
+Various experiments on multiple datasets show the efficacy of the method as well as out performing or showing comparable results to state-of-the-art

-In the caption for Table 1 the author’s write: “the high likelihood and lower difference between EAAD and MAAD indicate that the Bingham loss better captures the underlying noise.” How much difference between EAAD and MAAD is considered significant and why?

-In section 4.5 they write “While Von Mises performs better on the MAAD, we observe that there is a larger difference between the MAAD and EAAD values for the Von Mises distribution than the Bingham distribution. This indicates that the uncertainty estimates of the Von Mises distribution may be overconfident.” Same question as above. What amount of difference between MAAD and EAAD is considered significant and why?


**Experience Assessment:**

I do not know much about this area.

**Review Assessment: Checking Correctness Of Derivations And Theory:**

I assessed the sensibility of the derivations and theory.

**Review Assessment: Checking Correctness Of Experiments:**

I carefully checked the experiments.

**Review Assessment: Thoroughness In Paper Reading:**

I read the paper at least twice and used my best judgement in assessing the paper.

---

> ### Author Response · Authors · 2019-11-12
> **Response to Reviewer 3**
>
> Thank you for your review and for recognizing the contributions of our work. As noted in [LPB2017], calibration and accuracy are two orthogonal concepts and, thus, require independent evaluation. We used the difference between EAAD and MAAD to get some insight into the former that can also easily be interpreted. In practice, the acceptable difference between these two metrics is application dependant. For instance, in certain grasping applications being a few degrees wrong about the orientation of an object might be more acceptable than the same amount of error in motion estimation on autonomous vehicles.  We are adding a discussion about this to the experiments section of our paper.
>
> References
> [LPB2017] B. Lakshminarayanan A. Pritzel, and C. Blundell. Simple and Scalable Predictive Uncertainty Estimation using Deep Ensembles, NeurIPS 2017.

---

### Official Review · AnonReviewer2 · 2019-11-01
**Official Blind Review #2**

**Rating:** 6

**Review:**

This paper focuses on the problem of reasoning about uncertain poses and orientations. To address the limitations of current deep learning-based approaches, the authors propose a probabilistic deep learning model with a novel loss function, Bingham loss, to predict uncertain orientations. The experimental results demonstrate the effectiveness of the proposed approach.

This paper is well-motivated and the proposed method addresses important problems in uncertain orientation prediction. The paper is well-supported by theoretical analysis, however, the empirical analysis is a little weak and the model does not consider multimodal cases. For the above reasons, I tend to accept this paper but wouldn't mind rejecting it.

Questions:
1. You only compared with one baseline. How does the model compare with a loss that is not based on directional statistics or Gaussian models?
2. How can you improve the model for multimodal cases?

**Experience Assessment:**

I have read many papers in this area.

**Review Assessment: Checking Correctness Of Derivations And Theory:**

I assessed the sensibility of the derivations and theory.

**Review Assessment: Checking Correctness Of Experiments:**

I assessed the sensibility of the experiments.

**Review Assessment: Thoroughness In Paper Reading:**

I read the paper at least twice and used my best judgement in assessing the paper.

---

> ### Author Response · Authors · 2019-11-12
> **Response to Reviewer2**
>
> Thank you for considering our paper and recognizing its contribution. We address your questions below.
>
> 1. “You only compared with one baseline. How does the model compare with a loss that is not based on directional statistics or Gaussian models?”
>
> In the newly added evaluations we included two non-probabilistic baselines for UPNA and IDIAP : The first loss is based on using a mean square error on the difference between the ground-truth and predicted quaternion and the second is a cosine based loss on the biternion representation. We will also add non-probabilistic baselines for T-Less.
>
>
> 2. “How can you improve the model for multimodal cases?”
>
> Most techniques that are used for estimating multimodal Gaussians are also applicable for the Bingham case. We are currently running evaluations using a Bingham variant of Mixture Density Networks on the T-Less dataset. Analogously to the Gaussian case, this may fail when trained directly. Thus, we first pretrain the MDN assuming the dispersion parameter Z to be fixed. Then, we train the entire network jointly. Similar strategies are also usual for Gaussian MDNs (as e.g. for the MDN baselines in [MICB19]).
>
> References
> [MICB19] O. Makansi, E. Ilg, O. Cicek, and T. Brox, Overcoming Limitations of Mixture Density Networks: A Sampling and Fitting Framework for Multimodal Future Prediction, CVPR 2019.

---

### Public Comment · ~Tolga_Birdal3 · 2019-10-14
**Good idea. Please polish further though.**

This is indeed an important problem and the authors take an important direction. However, I would like to point out a couple of obvious issues that prevent this paper to be a good one:

1. The Gram-Schmidt (GS) orthonormalization process is unnecessary and I would like to discourage the authors and the community from following that path. I understand that today's auto-grad methods make it seamless to use such arbitrary complex functions. But in this case there are two issues: (a). If one were to differentiate the GS by hand one would get a good grasp of the complexity involved. This means that the network will train or run slower. (b). Even if the computational aspects are not a problem, GS is known to be numerically unstable and one usually ends up with vectors that are often not quite orthogonal. Modified-GS (not employed in this paper) is one way to overcome this, but there is no guarantee to avoid the numerical issues altogether.

Using the parallelizable nature of quaternions, in our NeurIPS 2018 work, we have already given an alternative way as a more elegant solution to construct the M matrix for quaternions and this doesn't suffer from numerical issues:  https://arxiv.org/pdf/1805.12279.pdf (look above equation 6). I suggest to employ that in future research.

2. The paper only deals with the single modal case. This is of some limited interest only. I would certainly like to see a better treatment of this problem, rather than a half done solution.
The paper claims that it deliberately avoids using "Mixture density networks" (MDN) for evaluation reasons. However, even if used, MDNs are known to suffer for such higher dimensional and complex multimodal distributions. One certainly needs cleverer strategies and I encourage the authors to continue to work on the problem, making it a strong contribution. It is not very convincing to just leave out the multimodal scenario.

---

> ### Author Response · Authors · 2019-11-12
> **Response to Public Comment**
>
> Thank you for appreciating our idea and for providing us with some interesting suggestions. We address your questions below.
>
> 1. Stability of Gram-Schmidt
> Both your approach and the Modified Gram-Schmidt (MGS) were added to our experiments. While we did not experience big differences between the Classical Gram-Schmidt (CGS) and MGS, we will add a discussion of numerical robustness to the paper acknowledging the quadratic dependency of CGS on the condition number of the input matrix [GLR05].
>
> Furthermore, we agree that whenever possible simpler neural network modules should be preferred over more complex ones. Particularly when the uncertainty is isotropic (i.e. the first three entries of Z are equal) considerable reductions of the output space are possible and the proposed matrix V can be readily used. As you write in your paper, V(q) is an injective mapping (from a 3 dimensional manifold) to the ring of orthonormal matrices (which is a 6d manifold [Lee03, Example 8.33]). Thus, by using V(q), we would restrict the expressiveness of our loss model.
>
> 2. Multi-Modal Representation
> We are incorporating a discussion of how our approach can be extended to Mixture Models and are currently evaluating Bingham Mixture Density Networks. See the response to the reviewer above for further details.
>
> References:
> [Lee03] J.M. Lee, Introduction to Smooth Manifolds, Springer, 2003.
>
> [GLR05] L. Giraud, J. Langou, M. Rozložník, and J. v. d. Eshof. Rounding error analysis of the classical Gram-Schmidt orthogonalization process, Numerische Mathematik 101(87), 2005.

---

### Author Response · Authors · 2019-11-15
**Uploaded Revised Version**

We would like to thank all reviewers again for their thoughtful feedback. We have uploaded another revised version of the paper. The changes focus on the aspects raised in the reviewers’ questions:

1. We address the main points of Reviewer 2 by including and discuss the training of mixture density networks as well as the incorporation of non-probabilistic baselines (Cosine and MSE loss) on the T-Less dataset. In our new experiments, we also demonstrate how the multi-stage training scheme is beneficial for the unimodal case when operating in a high uncertainty regime.
2. We address the points raised by Reviewer 3 in detail by explicitly investigating the role of MAAD and EAAD during training. First, we demonstrate how these metrics can provide insight into the size of the lookup table (we demonstrate how the EAAD may be higher than MAAD if the lookup table does not cover a sufficient range) and provide insights on choosing this range. Second, we further elaborate on approximation quality and overfitting.

Please feel free to get back to us if you have any further questions or feedback and we will be happy to incorporate it.

---

### Decision · Program_Chairs · 2019-12-19

**Decision:**

Accept (Poster)

**Comment:**

This paper considers the problem of reasoning about uncertain poses of objects in images. The reviewers agree that this is an interesting direction, and that the paper has interesting technical merit.